implementation research outcome; mental health; common mental disorder; task-sharing; Friendship Bench

**Corresponding author:**
John Patena;
Email: john.patena@nyu.edu

# Evaluating implementation research outcomes for a task-sharing mental health intervention: A systematic review of the Friendship Bench

John Patena ⬦, Deborah Adenikinju, Priyanka Lanka, Tania Hameed, Sumedha Kulkarni, Nana Osei-Tutu, Sophia Zuniga, Christina Ruan, Shivani Shenoy, Diksha Thakkar, Elizabeth Noble, Brian Angulo, Dorice Vieira, Joyce Gyamfi and Emmanuel Peprah

Implementing Sustainable Evidence-Based Interventions through Engagement (ISEE) Lab, Department of Global and Environmental Health, New York University School of Global Public Health, New York City, NY, USA

## Abstract

Common mental disorders (CMDs) are a leading cause of burden and disability globally. Approximately 75% of those living with CMDs reside in low- and middle-income countries (LMICs), and up to 90% of those needing mental health care do not receive it. The Friendship Bench is a task-sharing mental health intervention delivered by lay health workers (LHWs) that utilizes concepts of Problem-Solving Therapy. The aim of this systematic review is to identify and evaluate the barriers and facilitators to the implementation of research outcomes of the Friendship Bench and understand its systematic uptake to narrow the CMD treatment gap. We conducted a systematic review of articles that reported on the Friendship Bench in LMICs, CMDs, implementation research outcomes, and studies that utilized experimental, observational, or qualitative study designs. We identified articles using medical subject headings and keywords from APA PsycINFO, Cochrane, CINAHL, EMBASE, Global Health, OVID, PubMed/Medline, Science Direct, Web of Science, and Google Scholar in February 2023 and again in December 2023 to capture any additional articles. We screened 641 articles, and a total of 7 articles were included in the final analysis. All studies were conducted in Zimbabwe within the past 8 years, and across all the studies, all implementation research outcomes were reported. There is strong evidence that the Friendship Bench is acceptable, appropriate, and feasible to address the CMD treatment gap in Zimbabwe. Facilitators include that the Friendship Bench is culturally adaptable, utilizes trusted LHWs, and has relatively strong community and political buy-in. Conversely, barriers include a lack of a reliable mental health system, limitations in its ability to treat more serious mental conditions, and mental health stigma. There is an opportunity to explore the application of the Friendship Bench for CMDs in other countries and as a basis for novel task-sharing interventions for other health conditions.

## Impact statement

This systematic review is the first to explore the implementation research outcomes of the Friendship Bench to identify and evaluate the barriers and facilitators to its systematic uptake to narrow the mental health treatment gap for common mental disorders. The review highlights that there is high acceptability, appropriateness, feasibility, and scalability in Zimbabwe. There is an opportunity to understand the implementation efforts of the Friendship Bench in other countries. Although the Friendship Bench has been adapted for other health conditions (e.g., HIV), this is the first review to systematically explore implementation efforts specific to mental health conditions.

## Introduction

Depressive and anxiety disorders are collectively referred to as common mental disorders (CMDs), which impact the mood or feelings of affected persons. These diagnosable health conditions are termed "common" because of the relatively high prevalence in the population, with symptoms ranging in their severity and duration (Health NCCfM, 2011). Globally, depressive disorders are estimated to impact 4.4% of the population, and anxiety disorders are estimated at 3.6%, with both categories having higher prevalence among females (World Health Organization, 2017). CMDs are one of the leading causes of burden and disability globally, particularly when approximately 75% of those living with CMDs reside in low- and middle-income countries

(LMICs) (Moitra et al., 2022). The 2019 Global Burden of Disease Study ranked depressive disorders as the 13th leading cause of overall burden (Lancet, 2020). Furthermore, there is an alarming treatment gap for CMDs in LMICs. Up to 90% of those needing mental health care do not receive it, and among those who utilize mental health services do not receive adequate treatment (Patel et al., 2010; Alonso et al., 2018).

Global mental health efforts to address this treatment gap have utilized task-sharing strategies to expand mental health care and improve access to the most vulnerable populations (van Ginneken et al., 2013; Patel et al., 2018). Task-sharing is the redistribution of care normally provided by mental health specialists (e.g., psychologists, psychiatrists) to non-mental health specialists or those with no prior mental health training (e.g., lay/community health workers) (Raviola et al., 2019). While there is demonstrated evidence of the effectiveness of task-sharing mental health interventions to address CMDs in a variety of settings in LMICs (Karyotaki et al., 2022; Prina et al., 2023), there remains a need to understand how to implement and scale up these interventions broadly in all communities and countries that need more mental health services to address the treatment gap.

The Friendship Bench is a task-sharing mental health intervention that utilizes concepts of Problem-Solving Therapy (PST) to address CMDs. Developed from Cognitive Behavioral Therapy concepts, PST is a brief, step-by-step approach to cognitive problem solving that involves focusing on practical solutions and skill-building (Zhang et al., 2018). The Friendship Bench was first developed by Dr. Dixon Chibanda in Zimbabwe (Chibanda et al., 2011). The intervention is delivered through six sessions by lay health workers (LHWs) who are trained in a manualized script and supervised by a mental health professional in primary care facilities. LHWs include trusted elders in the community, with some being referred to as "grandmothers." The Friendship Bench intervention is delivered on a physical bench in a discreet area outside of a health facility (Chibanda et al., 2011; 2015; 2016b). In Zimbabwe, depressive symptoms are referred to as *kufungisisa* (the concept of "thinking too much" in Shona – a Bantu language widely spoken in Zimbabwe). Key components of the Friendship Bench include *kuvhura pfungwa* ("opening of the mind"), *kusimudzira* ("uplifting") and *kusimbisa* ("strengthening"). The full description of the intervention and how it was developed is described elsewhere (Chibanda et al., 2011; 2015; 2016b).

Since it was first developed in 2006, the Friendship Bench has been scaled up in over 100 urban health facilities in Zimbabwe. Improvements in CMD symptoms were demonstrated and measured by a reduction in the Shona Symptom Questionnaire, a locally validated screening tool for CMDs (Chibanda et al., 2016a; 2016d; Chibanda, 2017). One study has evaluated its reach (Verhey et al., 2022), with several more exploring areas for future application in addressing anxiety (Abas et al., 2020), trauma (Verhey et al., 2020; 2021), suicide (Munetsi et al., 2018) and rural areas in Zimbabwe (Fernando et al., 2021; Brown et al., 2022). The Friendship Bench has also been formatively adapted to a variety of other health conditions including: people living with HIV and adherence to antiretroviral therapy (Chinoda et al., 2020; Stockton et al., 2020; Haas et al., 2021; Ouansafi et al., 2021; Simms et al., 2021; Stockton et al., 2021; Wogrin et al., 2021; Simms et al., 2022; Bengtson et al., 2023; Garriott et al., 2023; Haas et al., 2023); people living with HIV and on methadone maintenance (Tran et al., 2022); noncommunicable diseases (Kamvura et al., 2021; 2022); youth in need of mental health care – aptly named the "Youth Friendship Bench" (Brostrom et al., 2021; Wallén et al., 2021; Brooks et al.,

2022); and a digital application of the Friendship Bench named "Inuka" (Dambi et al., 2022; Doukani et al., 2023). The Friendship Bench has been extensively researched in Zimbabwe, with other research conducted in Botswana (Brooks et al., 2022; Garriott et al., 2023), Kenya (Doukani et al., 2023), Malawi (Stockton et al., 2020; 2021; Bengtson et al., 2023) and Vietnam (Tran et al., 2022). While the Friendship Bench can act as a tool to narrow the treatment gap for CMDs in LMICs, an understanding of its implementation determinants (i.e., barriers and facilitators) and successes (i.e., implementation outcomes) is needed to inform scale-up of the Friendship Bench in other LMICs.

To our knowledge, no systematic review has ever been conducted on the Friendship Bench, which highlights the importance and timely manner of this investigation. Therefore, the aim of this systematic review is to identify and evaluate the implementation research outcomes of the Friendship Bench and understand the barriers and facilitators to its systematic uptake to narrow the CMD treatment gap.

Proctor et al.'s (2011) implementation research outcomes taxonomy has defined how the implementation science field conceptualizes and evaluates implementation success: acceptability, adoption, appropriateness, costs, feasibility, fidelity, penetration, and sustainability (Proctor et al., 2011). A recent scoping review by Proctor et al. (2023) highlighted that the term "scaling up" emerged as a new concept over the past 10 years (Proctor et al., 2023). Scalability, defined as efforts to increase the impact of an intervention widely, posits that there can be a guided process for maximum implementation of an intervention (Zamboni et al., 2019). Implementation researchers have advocated the importance of scalability/scaling-up as a needed measurement for effective uptake and sustainability (Gyamfi et al., 2021; 2022).

A comprehensive systematic review by Le et al. (2022) developed the "Barriers and Facilitators in Implementation of Task-Sharing Mental Health Interventions" (BeFITS-MH) conceptual framework which is comprised of 37 constructs across eight domains: (1) patient/client characteristics, (2) provider characteristics, (3) family and community factors, (4) organizational characteristics, (5) societal factors, (6) mental health system factors, (7) intervention characteristics and (8) stigma (Le et al., 2022). The factors most amenable to change were most cited as facilitators, including intervention characteristics (i.e., setting, format) and provider characteristics (i.e., knowledge, skills). Conversely, barriers consisted of factors at the macro-level, including societal factors (i.e., sociocultural norms, economic conditions) and stigma to mental illness (Le et al., 2022). The BeFITS-MH framework highlights an ongoing challenge in global mental health efforts that calls for more implementation strategies to integrate task-sharing mental health interventions across a variety of settings.

## Methods

### Search strategy

We developed a comprehensive search strategy to identify published articles that met predefined inclusion criteria using the Preferred Reporting Items for Systematic reviews and Meta-Analysis (PRISMA) (Appendix 1, Supplementary Material – PRISMA checklist) (Moher et al., 2009). The World Bank criteria in 2023 was used to define LMICs (The World Bank, 2023). We identified articles using medical subject headings and keywords including "common mental disorders," "mental health," "Friendship Bench," "LMIC", and all implementation research outcome terms (Appendix 2,

Supplementary Material – Search strategy). We searched the following databases: APA PsycINFO, Cochrane, CINAHL, EMBASE, Global Health, OVID, PubMed/Medline, Science Direct, Web of Science, and gray literature (Google Scholar). The article search was initially conducted in February 2023 and ran again in December 2023 to capture any additional articles. This systematic review was registered on the Open Science Framework on February 10, 2023 (https://doi.org/10.17605/OSF.IO/D8PE7).

### Inclusion and exclusion criteria

Articles were included if they met the following inclusion criteria: (1) reported on the Friendship Bench in LMICs, (2) reported on CMDs, (3) reported on implementation research outcomes defined by Proctor et al. (2011), and (4) reported using experimental, observational, or qualitative study designs, including case studies. Protocols, commentaries, reviews of any type, and studies that reported on the Friendship Bench as a model for other health outcomes (e.g., antiretroviral adherence for HIV care) were excluded. There were no restrictions on publication date or language.

### Data extraction

All citations were downloaded to Covidence. Titles and abstracts of all articles were independently screened and rated by two reviewers to determine if they met inclusion criteria (JP, NO, SZ, CR, SS, DT, EN, BA). Discrepancies were resolved by consensus. A full-text article review was then conducted, and relevant information was extracted by two reviewers (JP, PL, SK, NO, SZ, CR, SS, DT, EN, BA). Specifically, the following study characteristics were retrieved and coded: study location, study design, description of how the Friendship Bench was implemented, implementation research outcome, and barriers and facilitators to implementation. All data were extracted and stored in Covidence.

### Quality assessment

To assess the risk of bias, "A comprehenSive tool to Support rEporting and critical appraiSal of qualitative, quantitative and mixed methods implementation reSearch outcomes" (ASSESS Tool) was utilized (Ryan et al., 2022). The 24-item tool helps to standardize the synthesis and reporting of implementation efforts and to describe studies evaluating implementation research outcomes. After selecting the study design, five questions use criterion to evaluate the design. As an example, qualitative studies use the following criteria: (1) Is the qualitative approach appropriate to answer the research question?; (2) Are the qualitative data collection methods adequate to address the research question?; (3) Are the findings adequately derived from the data?; (4) Is the interpretation of results sufficiently substantiated by data?; and (5) Is there coherence between qualitative data sources, collection, analysis, and interpretation? Criteria differ for each kind of study design. Each question receives a binary score to indicate whether each criterion was met (1) or not met (0). After summing scores across the five criteria, the risk of bias is categorized as high bias (score of 1–2), low bias (score of 3–5), or unclear (unable to be assessed). Two reviewers independently assessed each article and rated for risk of bias (JP and TH).

## Results

A total of 685 articles were identified. After removing duplicates, 641 titles and abstracts were screened. Of those, 595 were excluded, yielding 46 articles for which full texts were obtained and reviewed. During the full text review and data extraction, 39 were excluded for wrong study design, wrong intervention, or no implementation outcome. Thus, only 7 articles met all study inclusion criteria and were included for the final analysis (Figure 1).

### Study characteristics

All seven included articles were conducted in Zimbabwe and published within the past 8 years of this systematic review (Abas et al., 2016; Chibanda et al., 2016c; 2016d; Chibanda, 2017; Fernando et al., 2021; Healey et al., 2022; Verhey et al., 2022). More than half of the studies (n = 4) utilized qualitative study design components, including focus group discussions and in-depth, semi-structured interviews with either patients, LHWs, or community stakeholders as either a standalone qualitative study (Fernando et al., 2021), case study (Chibanda, 2017) or part of a mixed-methods analysis (Abas et al., 2016; Chibanda et al., 2016c). Quantitative, non-randomized study designs components (n = 4) included: conducting descriptive analyses of routine intervention attendance data (Abas et al., 2016) and a needs assessment and skills assessment (Chibanda et al., 2016c) – both as part of a mixed-methods analysis; an economic threshold analysis which used a modeling-based deterministic threshold analysis using a cost-utility framework (Healey et al., 2022); and evaluating implementation reach, adoption, and implementation by developing indicators using the "Reach, Efficacy, Adoption, Implementation, Maintenance Framework" (RE-AIM) (Verhey et al., 2022) – a widely used implementation science framework to evaluate outcomes on the process of scaling up evidence-based interventions (Gaglio et al., 2013). Only one study utilized a clustered randomized control trial with the Friendship Bench as the intervention and enhanced care as control (Chibanda et al., 2016d). The study characteristics are provided in Table 1.

### Implementation of research outcomes

All included articles were assessed on the explicit reporting or description of the concept for the following implementation research outcomes: acceptability, adoption, appropriateness, costs, feasibility, fidelity, penetration, and sustainability, with additional attention to scalability or "scaling-up". Of the studies included, three reported on acceptability (Abas et al., 2016; Chibanda et al., 2016c; Fernando et al., 2021), one reported on adoption (Verhey et al., 2022), two reported on appropriateness (Abas et al., 2016; Chibanda et al., 2016c), one reported on costs (Healey et al., 2022), one reported on feasibility (Chibanda et al., 2016c), two reported on fidelity (Chibanda et al., 2016d; Chibanda, 2017), one reported on penetration (Verhey et al., 2022) and two reported on sustainability (Fernando et al., 2021; Healey et al., 2022). Even though scalability is not part of the original Proctor et al. (2011) taxonomy (Proctor et al., 2011), it was mentioned or described in all but one study, highlighting the significance of its consideration as an implementation outcome.

Acceptability – the perception that an evidence-based intervention is agreeable or satisfactory – constitutes a wide range from explicitly being reported as the number of patients who utilized the Friendship Bench, workforce retention and themes from qualitative feedback (Abas et al., 2016; Chibanda et al., 2016c; Fernando et al.,

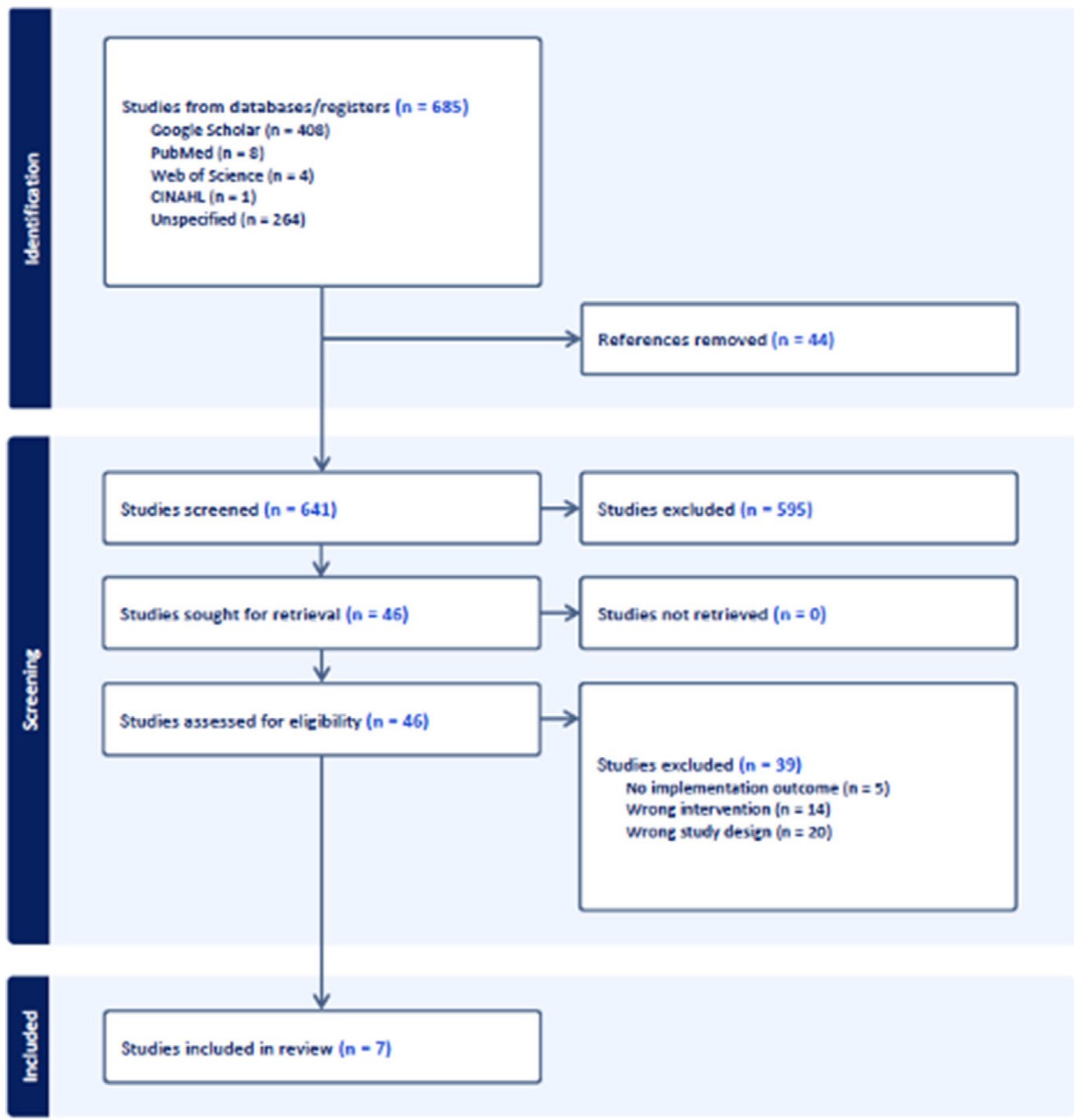

**Figure 1.** PRISMA flow diagram.

2021). One study that implemented the Friendship Bench in Zimbabwe reported that between 2010 and 2014, there were 5,434 total visits, averaging 505 per year, including multiple visits. A small percentage (5.7%) of patients received 4 or more sessions. In those 4 years, 14 out of 15 LHWs continued to deliver the Friendship Bench, demonstrating strong workforce retention. The study reports that these two statistics are an indicator of high acceptability (Abas et al., 2016). Patient feedback from focus groups and in-depth interviews found that the Friendship Bench had high socio-cultural acceptability because (1) the intervention used local terminology for emotional distress (*kufungisisa* – the concept of "thinking too much" in Shona) instead of using stigmatizing terms

such as depression; (2) the intervention focused on identifying problems and behavioral activation (what to do about it) – as opposed to identifying their challenges as "feelings"; and (3) the perception that LHWs were like family – in some contexts as "grandmothers" – who shared similar social and economic problems as the patients, perceiving the LHWs as more relatable and more suitable to deliver the intervention (Abas et al., 2016; Chibanda et al. 2016c; Fernando et al., 2021). Patients welcomed the Friendship Bench as a way to "open up the mind" and expressed wanting the intervention to keep going beyond the initial research study (Abas et al., 2016; Chibanda et al., 2016c; Fernando et al., 2021). LHWs reported that the Friendship Bench was

**Table 1.** Characteristics of the studies included in the systematic review

| Author, Year | Country | Aim | Study Design | Acceptability | Adoption | Appropriateness | Cost | Feasibility | Fidelity | Penetration | Sustainability | Scalability |
|---|---|---|---|---|---|---|---|---|---|---|---|---|
| Abas et al. (2016) | Zimbabwe | To investigate the FB's acceptability, understand the facilitators and challenges to implementation, and inform ways to scale-up the FB across low-income settings. | Mixed-methods: descriptive analyses on routine attendance data; focus group discussions with LHWs; in-depth interviews with clinic staff and patients | X number of patients and visits; workforce turnover; themes that emerged from focus group discussions and in-depth interviews | | X number of patients and visits; workforce turnover; themes that emerged from focus group discussions and in-depth interviews | | | | | | X number of patients and visits; workforce turnover; themes that emerged from focus group discussions and in-depth interviews |
| Chibanda et al. (2016a) | Zimbabwe | To evaluate the effectiveness of a culturally adapted psychological intervention for CMDs delivered by LHWs in primary care. | RCT: cluster randomized control trial with 6-month follow-up – clinics randomized to FB or enhanced usual care (control) | | | | | X delivering FB across all sites | | | | X designing an intervention delivered within the health system and using existing workers |
| Chibanda et al. (2016b) | Zimbabwe | Describe the process taken by key stakeholders to ensure that a scale-up plan of the FB was acceptable and feasible. | Mixed-methods: needs assessment; skills assessment; consultation workshops; in-depth interviews/ consultations | X stakeholder feedback addressing several layers of engagement | | X stakeholder feedback addressing several layers of engagement | | X stakeholder feedback addressing several layers of engagement | | | | X stakeholder feedback addressing several layers of engagement |
| Chibanda (2017) | Zimbabwe | Describe the three-pronged approach that led to the scale-up of the FB. | Qualitative: case study describing approach to implementation | | | | | X delivering FB across multiple health facilities | | | | X delivering FB across multiple health facilities |
| Fernando et al. (2021) | Zimbabwe | To evaluate the FB delivered by village health workers in rural Zimbabwe. | Qualitative: semi-structured interviews with participants | X themes that emerged from qualitative interviews | | | | | | | X themes that emerged from qualitative interviews | |

(*Continued*)

**Table 1.** (*Continued*)

| Author, Year | Country | Aim | Study Design | Acceptability | Adoption | Appropriateness | Cost | Feasibility | Fidelity | Penetration | Sustainability | Scalability |
|---|---|---|---|---|---|---|---|---|---|---|---|---|
| Healey et al. (2022) | Zimbabwe | To assess the level of treatment coverage needed for a scale-up of the FB to be considered a cost-effective investment. | Quantitative, non-randomized: economic threshold analysis conducted within a "cost-utility" framework | | | | X results and interpretation from economic threshold analysis | | | | X results and interpretation from economic threshold analysis | X results and interpretation from economic threshold analysis |
| Verhey et al. (2022) | Zimbabwe | To evaluate the implementation of the FB using the RE-AIM framework. | Quantitative, non-randomized: developed indicators using RE-AIM framework; descriptive and inferential statistics | | X results from indicators organized by Reach, Adoption, and Implementation domains | | | | | X results from indicators organized by Reach, Adoption, and Implementation domains | | X results from indicators organized by Reach, Adoption, and Implementation domains |

patient-centered, allowing patients to drive the direction of how they wanted to be helped, while reporting a sense of personal reward and satisfaction being an "agent of change" (Abas et al., 2016).

Adoption – the intentional action to employ an evidence-based intervention, sometimes referred to as "uptake" – was one of the main, explicitly named indicators created in the RE-AIM study (Verhey et al., 2022). Sources for the adoption indicator included the number of community health workers who attended a Friendship Bench training, the number of supervisors available, the number of participants seen per month, and whether the clinic had the bench installed. The aggregate scores indicated high adoption overall, ranging from 59% (small clinics) to 71% (large clinics) (Verhey et al., 2022).

Appropriateness – the perceived fit of the evidence-based intervention for a setting, provider, or consumer – while not explicitly reported on, was captured broadly and jointly with acceptability (Abas et al., 2016; Chibanda et al., 2016c). It can be inferred from studies that discuss patient perceptions that the Friendship Bench helped improve their problems of "thinking too much," feel more valued in their community, and feel less stigmatized. Feedback from LHWs highlights that the Friendship Bench has a supportive supervision structure and that being trained in PST supported their own lives and the challenges they were experiencing (Abas et al., 2016; Chibanda et al., 2016c).

Cost was reported as the level of treatment coverage needed for the scale-up of the Friendship Bench to be considered cost-effective (Healey et al., 2022). A modeling-based deterministic threshold analysis was conducted within a "cost-utility" framework. The authors estimated that an additional 3,413 "service users" (participants who would use the Friendship Bench) would need to be treated each year, estimated to be an additional 10 service users per known active LHW. The associated incremental cost-effective ratio was $191 per year lived with disability avoided. The study concluded that this calculation was reasonable and a convincing reason to scale up the Friendship Bench (Healey et al., 2022).

Feasibility – the extent to which an evidence-based intervention can be successfully used or implemented – was reported broadly in terms of the Friendship Bench being "feasible to implement" from stakeholder feedback through qualitative interviews. It was reported jointly with acceptability, emphasizing that ongoing training sessions and securing funding from capacity-building grants help with its feasibility (Chibanda et al., 2016c).

Fidelity – the extent to which an evidence-based intervention is implemented as it was originally intended or created – was reported to be a challenge, especially when outside the contexts of a clinical trial (Chibanda et al. 2016c; Chibanda, 2017). One study describes communication challenges between LHWs and their supervisors, along with the hope that using mobile phone features such as texting (e.g., short-message-service, WhatsApp) and communication management tools (e.g., Slack) will mitigate the issue (Chibanda, 2017). The main paper that assessed overall Friendship Bench effectiveness describes audio recording sessions to ensure adherence to the intervention (Chibanda et al., 2016d). Both studies emphasize the need to investigate fidelity more as the Friendship Bench scales up (Chibanda et al., 2016d; Chibanda, 2017).

Penetration – the integration of an evidence-based intervention within a service setting – was not explicitly reported, though its concept was reported as "reach" in the study using the RE-AIM framework (Verhey et al., 2022). The reach indicator was calculated as the percent of participants registered at the clinic receiving the Shona Symptom Questionnaire, and of those who met a screening threshold, the percent of participants who received the Friendship Bench intervention. Small clinics achieved 34% reach, medium clinics at 9% reach and large clinics at 15% reach, which the authors reported as overall low (Verhey et al., 2022).

Sustainability – the extent to which an evidence-based intervention is maintained or institutionalized within a setting – was reported by one study as the percentage of patients still engaged with the Friendship Bench after 1 year of the initial research study (Fernando et al., 2021). Fifty-two percent of patients were still actively engaged and reported that the Friendship Bench continued to "relax their mind." Additionally, 67% maintained or expanded their income-generating projects as a result of improved mental health, a factor that the study explained as a positive long-term outcome (Fernando et al., 2021). The economic analysis threshold described in *Costs* advocates that the Friendship Bench has the potential to be a cost-effective intervention, which supports long-term sustainability (Healey et al., 2022).

Scalability – the efforts to increase the impact of an evidence-based intervention widely – was mentioned explicitly in all but one of the studies, highlighting the importance of "scaling up" the Friendship Bench (Abas et al., 2016; Chibanda et al., 2016c; 2016d; Chibanda, 2017; Healey et al., 2022; Verhey et al., 2022). Themes that emerged included (1) needing to integrate the intervention within existing health systems and workflows (Chibanda et al., 2016d; Verhey et al., 2022); (2) utilizing LHWs with their existing competencies and connections with the community (Abas et al., 2016; Chibanda et al., 2016c); (3) maximizing community engagement by involving stakeholders as partners and focusing on building trust (Abas et al., 2016; Chibanda, 2017); (4) political buy-in from local health authorities (Chibanda et al., 2016c; Chibanda, 2017); and (5) demonstrating how cost-effective the intervention is (Healey et al., 2022).

### Facilitators to implementing the Friendship Bench

Patient/client characteristics – The most relevant personal attributes that facilitated implementation success were a patient's motivation or readiness to participate in the Friendship Bench. The evidence on acceptability and appropriateness highlights that patients were eager to try the Friendship Bench. As illustrated by the BeFITS-MH framework, this is the most cited facilitator and, as a result, predictor for success. While a patient's baseline skills and self-efficacy for help-seeking behaviors are also important facilitators in this domain, it was not explicitly measured in the included studies.

Task-sharing provider characteristics – The personal attributes of the task-sharing provider (e.g., LHWs) were also highly cited as facilitators to implementation. LHWs' skills, self-efficacy, and knowledge were measured in the skills assessment and showed high levels of competency in being approachable, trustworthy, mature, "motherly," and having listening skills. Additionally, LHWs' other personal attributes, being perceived as family members, their trusted role in the community, and being perceived as having the same social and economic problems as the patients all contribute to facilitation. LHWs are deeply integrated in the community and have a deep sense of local norms, culture, context, and understanding of the issues faced by the community. The social role and identity that LHWs perceived themselves to be "agents of change" also contributes significantly (Abas et al., 2016; Chibanda, 2017). The BeFITS-MH framework highlights these attributes as top facilitators in this domain and therefore leads to increased acceptability of the intervention.

Intervention characteristics – Unsurprisingly, components of the Friendship Bench intervention itself were overwhelmingly discussed as facilitators. From using local terminology to describe emotional distress, to focusing on "problems" as opposed to "feelings," to having a patient-centered approach allowing the patient to be the driver of their treatment, the Friendship Bench resonated positively for both patients and LHWs. The supportive supervision structure – having access to peer LHWs and mental health professionals – led to the top-cited LHW perspective of feeling well-supported. Using the Friendship Bench as a separate entity to healthcare facilities – dyads meet in a discreet area on a physical bench – was also a facilitator. The BeFITS-MH framework reports that these intervention characteristics are the most amenable to change: having the most agency in changing components of the intervention that will support the intended population. These factors lead to the acceptability of the components of the Friendship Bench.

Organizational factors – Broader factors at the organizational level include collaboration across community organizations and health authorities. Included studies described how there was governmental buy-in for the Friendship Bench when it was presented as (1) emphasizing the added value of treating CMDs as a comorbid condition to existing public health programs (e.g., HIV, malaria, non-communicable diseases), (2) emphasizing utilizing existing resources of LHWs and other infrastructure resources, and (3) when policymakers were invited as part of the stakeholder engagement process. These factors lead to the feasibility of implementing the Friendship Bench.

### Barriers to implementing the Friendship Bench

Intervention characteristics – While an extensive list of Friendship Bench intervention characteristics served as facilitators, there were also key barriers to successful implementation. LHWs overwhelmingly noted that there was insufficient training for assessing and managing more serious mental health conditions such as suicide, domestic violence, and hostile patients. There is also a lack of comprehensive documentation and follow-up, making it challenging to monitor who has received the intervention. This highlights key barriers in task-sharing mental health interventions in general and asks the question: is task-sharing "enough"? One perspective advocates that task-sharing mental health interventions provide help for those experiencing mild to moderate CMD symptoms (Karyotaki et al., 2022; Prina et al., 2023). Those that may require more comprehensive care, including pharmacological treatments and therapy, would not receive that level of care from task-sharing. Additionally, although the Friendship Bench is intended for a low-intensity psychological intervention, there could be an opportunity to expand to other mental and behavioral health conditions, such as post-traumatic stress disorder and substance use disorders.

Mental health system factors – Unsurprisingly, the infrastructure-related factors of the mental health system contribute significantly to barriers. Similar to the barriers noted above, if patients require more comprehensive services in primary care or medication management to address their mental health conditions, the Friendship Bench is limited to what it can offer. Included studies highlight that the overall lack of mental health training and resources, unreliable referral systems and lack of mental health professionals perpetuate the mental health treatment gap. The BeFITS-MH model reinforces these concepts by illustrating that the human resources needed for sufficient mental health care is deficient and task-sharing mental health interventions cannot be the only source of support for populations who need it. To address these issues, a coordinated effort among local health authorities and communities needs to be conducted. Establishing referral systems and capacity for mental health facilities will strengthen the task-sharing model.

Stigma – Perhaps the most complicated and complex factor in implementation, stigma is solely reported as hindering implementation success and cuts across all domains and levels of the BeFITS-MH framework (Le et al., 2022). Mental illness stigma in LMICs has been an ongoing challenge in the mental health treatment gap (Mascayano et al., 2015). Termed as a "universal phenomenon," stigma is perhaps the strongest barrier as it originates from personal, cultural, and societal mindsets of what "mental health" is. The Friendship Bench, while noting its many positive attributes, is still not immune to some populations' hesitancy to engage in an intervention that discusses personal problems and experiences.

### Quality assessment

The ASSESS Tool was used to determine the risk of bias based on the type of study design for each included article: mixed-methods, qualitative, quantitative, non-randomized and randomized control trial. The risk of bias is categorized as high bias (score of 1–2), low bias (score of 3–5), or unclear (unable to be assessed). Six of the included articles were rated between a score of 4 and 5, thus having low bias – an indicator of being a high-quality study. The remaining article, Chibanda et al.'s (Chibanda, 2017) case study (Chibanda, 2017), was deemed as unable to be assessed, given there was no epidemiologic study design used (Figure 2).

### Discussion

To our knowledge, this is the first systematic review to synthesize the findings on the implementation determinants and outcomes of the Friendship Bench as a task-sharing intervention to address common mental disorders. While LMICs were included in the search strategy, results only featured work in Zimbabwe, highlighting the implementation outcomes associated with the Friendship Bench for this setting only.

The seven articles included in this review reported on all implementation research outcomes per Proctor et al.'s (2011) taxonomy (Proctor et al., 2011) – acceptability ($n = 3$), adoption ($n = 1$), appropriateness ($n = 2$), costs ($n = 1$), feasibility ($n = 1$), fidelity ($n = 2$), penetration ($n = 1$), and sustainability ($n = 2$) – with an additional focus on scalability ($n = 6$), a burgeoning implementation research outcome that deserves attention. While included studies reported on the full taxonomy list, some are reported more explicitly while others are inferred. Acceptability is the most reported on (Abas et al., 2016; Chibanda et al., 2016c; Fernando et al., 2021), and clearly defined in studies on how it was measured and evaluated. Acceptability is one of the most widely evaluated implementation outcomes (Proctor et al., 2023) and is one of the most common questions to ask when implementing an intervention: how acceptable is this to the intended population? Similarly, adoption was explicitly created as an implementation indicator (Verhey et al., 2022) and cost is a straightforward outcome to measure (Healey et al., 2022). On the other hand, appropriateness and feasibility were not outcomes that were explicitly reported on or measured, but could be inferred based on how it was described in the studies; interestingly both were broadly jointed with acceptability (Abas et al., 2016; Chibanda et al., 2016c). While the term "penetration" was not explicitly reported on, "reach" is an accurate measurement

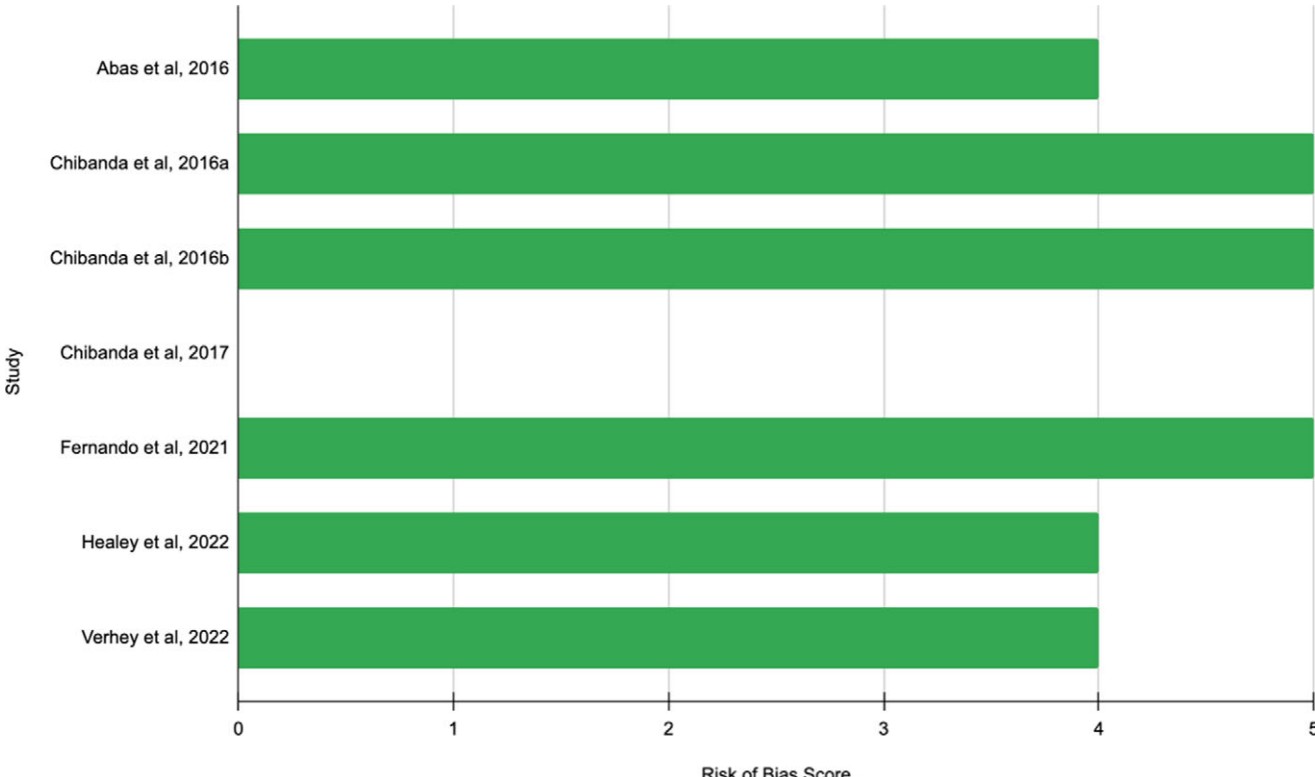

**Figure 2.** Risk of bias.

of the level of integration within a service setting and was formally evaluated using the RE-AIM framework (Verhey et al., 2022). Sustainability was also reported on by name, emphasizing the need to maintain the implementation of the Friendship Bench beyond the research study (Fernando et al., 2021; Healey et al., 2022). Overall, there is high acceptability, appropriateness, and feasibility of the Friendship Bench, with lower evidence demonstrating its overall reach and sustainability. To enhance the measurement and evaluation of these implementation research outcomes, studies could employ validated and reliable measures. Mettert et al. (2020) published a systematic review that identified 102 measures that could be used to quantitatively assess implementation outcomes for mental health and behavioral health studies (Mettert et al., 2020). None of the included studies for this review on the Friendship Bench utilized these measures.

Interestingly, although the Friendship Bench has been formatively researched in Botswana (Brooks et al., 2022; Garriott et al., 2023), Kenya (Doukani et al., 2023), Malawi (Stockton et al., 2020; 2021; Bengtson et al., 2023) and Vietnam (Tran et al., 2022), all included studies were in Zimbabwe because studies from these other locations address adaptations to the Friendship Bench for other health conditions. This review focused on the implementation research outcomes of the Friendship Bench as it addresses CMDs. Moreover, beyond research, the Friendship Bench has established itself as a non-governmental organization (www.friendshipbenchzimbabwe.org) registered in Zimbabwe as a Private Voluntary Organization with the Department of Social Service. The Friendship Bench organization provides the intervention as a manualized toolkit that can be implemented globally. According to their 2023 Annual Impact Report, the Friendship Bench has been implemented in Zimbabwe, Jordan,

Kenya, Malawi, Tanzania, the United Kingdom, the United States of America, and Vietnam to over 482,000 people delivered by 2,000 LHWs since 2016 (Bench, 2023). However, there is no published research on these implementation efforts.

This lack of reporting on implementation research outcomes highlights a challenge in the field: there are not enough studies being conducted to explicitly examine the full taxonomy of implementation outcomes. Only 1 of the 7 studies used an implementation science framework to formally evaluate an implementation outcome – the Verhey et al. (2022) study which utilized RE-AIM (Verhey et al., 2022). As Proctor et al. (2023) explained, implementation researchers are not as equipped or prepared to achieve implementation success when there is not enough research on outcomes such as penetration, sustainability or appropriateness using rigorous, analytical study designs (Proctor et al., 2023).

### Strengths and limitations

This study used a rigorous search strategy. Pre-specified inclusion and exclusion criteria were used to retrieve articles across multiple databases. No restrictions were placed on article publication date or language published in order to capture all relevant articles. A limitation included the exclusive focus on the Friendship Bench as it relates to addressing CMDs. There were several articles discussing implementation efforts; however, as they related to other health conditions, it was excluded from this review. Information from these other studies could have also contributed to understanding the Friendship Bench's scalability, especially since it was conducted in countries outside of Zimbabwe.

## Conclusion

This systematic review examined the implementation efforts of the Friendship Bench to address CMDs in LMICs. There is strong evidence that the Friendship Bench is acceptable, appropriate, and feasible to address the CMD treatment gap in Zimbabwe. Facilitators include that the Friendship Bench is culturally adaptable, utilizes trusted LHWs, and has relatively strong community and political buy-in. Conversely, barriers include a lack of a reliable mental health system, limitations in its ability to treat more serious mental conditions, and mental health stigma. The presented information on the facilitators and barriers is limited to the context of Zimbabwe. There is an opportunity to explore the application of the Friendship Bench for CMDs in other countries. Additionally, there is an opportunity to evaluate the implementation outcomes of the Friendship Bench as a basis for novel task-sharing interventions for other health conditions. The evidence demonstrates that the Friendship Bench holds promise for bridging the mental health treatment gap, however, more research is required.

**Open peer review.** To view the open peer review materials for this article, please visit http://doi.org/10.1017/gmh.2025.10025.

**Supplementary material.** The supplementary material for this article can be found at http://doi.org/10.1017/gmh.2025.10025.

**Data availability statement.** Data sharing is not applicable – no new data generated.

**Author contribution.** All authors contributed to the development of this manuscript. JP: conceptualization, formal analysis, project administration, writing – original draft. DA: formal analysis, writing – review & editing. PL: formal analysis, writing – review & editing. TH: formal analysis, writing – review & editing. SK: formal analysis, writing – review & editing. DT: formal analysis, writing – review & editing. EN: formal analysis, writing – review & editing. BA: formal analysis, writing – review & editing. DV: data curation, writing – review & editing. JG: writing – review & editing. EP: supervision, writing – review & editing.

**Financial support.** This research received no specific grant from any funding agency, commercial or not-for-profit sectors.

**Competing interests.** The authors declare none.

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
