## [Reviewer Report]

Page 3: Line 2: The impact statement is not adequately precise to give a clear understanding if what the systematic review is trying to achieve. It seems to be

Page 3: Line 6: Unclear what is meant by “opportunity to understand implementation efforts in other countries” as there is no qualifying sentence.

Page 3: Lines 17-19: The aim includes understanding its systematic uptake in LMIC - I don’t understand how this is possible through a systematic review! This would require an examination of the policy landscape in LMICS and if Friendship Bench (FB) is recognized as a model of intervention for CMDs in that country.

Page 3: Lines 21-25: The inclusion of study design criteria of the review as part of the abstract appears to be misplaced.

Page 4: Line 27: Unclear what is meant by “reported on all implementation research outcomes”

Page 5: Lines 48-50: Sentence is contradictory - if CMDs are a leading cause of burden and disability, why is it peculiar to LMICs?

References are cited inconsistently throughout!

Page 6: Line 67: FB is described as supporting “a more empowering approach toward identifying problems and finding workable solutions” - it would be helpful to know what about the FB promotes greater empowerment (which is not defined or explained)..

Page 6-7: Lines 80-98 - this section would perhaps be better integrated with the review section as it is a reflection of the various countries in which the FB has been used to study various problems!

Page 7: Lines 99-101: The statement cannot be true as there is no evidence that the the FB has narrowed the treatment gap in LMICs! In any case, perhaps this should be argued in the discussion section following the review.

Over half the studies that were accessed (n=264) indicates and unspecified database. Could the authors provide more information about these unspecified sources?

It is unclear how the denominator was calculated for each of Proctor’s implementation outcomes as each of these elements was not present in all studies. The authors should clarify how scores were aggregated in the reviewed studies.

Page 12: Line:224: A total of 5434 visits over 4 years is averaged at 505 per year - please indicate how this averaged was arrived at? Does this total include multiple sessions?

Page 12: Lines 225-227 Could the authors comment on if LHWs were paid or had other sources of income besides the work they did as this may either strengthen or weaken the evidence!

Page 13: Lines 246-248: It would be helpful if the numerator and denominator for adoption was reflected to make an appropriate judgement of the level of adoption. Similarly, where possible these figures should be reflected.

Could the authors explain how in some instances quantitative data was used and in other instances, greater reliance was placed on qualitative data to arrive at how well these studies satisfied Proctor’s criteria of implementation outputs. For example, on page 13: Lines 277-279, the authors use a recommendation from the main paper that assessed overall FB effectiveness to investigate fidelity more as “evidence” of fidelity.

Could the authors further comment on their evaluation that six of the seven studies had low bias, given that most of these were self-reports, perceptions and used qualitative methods with no comparison groups.

Overall: A major weakness of this review is that most, if not all the studies, were undertaken in Zimbabwe. The self-imposed limitation to only look at FB in relation CMDs reduces it applicability to other contexts. The title is therefore misleading as it does not cover LMICs in general. It may have been advisable to focus this review on Zimbabwe itself. In that vein, Regionally, the Zimbabwean health care system should be described more integrally for these findings to be extended to other health systems with a similar structure or resource constraints.

As such, its potential as a viable task-shared intervention in other countries or more globally is unknown.

---

## [Reviewer Report]

Thank you for the opportunity to review this article, which sought to systematically review the implementation efforts of the Friendship Bench as task-sharing mental health intervention model for common mental disorders (CMDs) in LMICs.

The article provides good background information of Friendship Bench, and a discussion of the implementation outcomes reported by the included articles. However, the entire article would benefit from a major re-structuring, to streamline its focus.

Currently, the title focuses on “implementation research outcomes” but the explicitly stated aim of the systematic review (in Abstract) is “to identify and evaluate the barriers and facilitators to the implementation of the Friendship Bench and understand its systematic uptake in LMICs to narrow the CMD treatment gap”. This dual focus (on both implementation outcomes and barriers and facilitators) makes the article disjointed, especially because the authors currently do not include a thorough analysis of the connections between barriers/facilitators and implementation outcomes -- just “implementation success”. For example, in the Results > Facilitators section, the authors discuss how patient/client characteristics have been linked to the implementation outcomes of acceptability and appropriateness, and provider characteristics to acceptability. However, no other linkages were mentioned. This might be a worthwhile exercise to explore.

Additionally, despite the aspirational goal of understanding scalability of Friendship Bench in other LMICs, all includes articles were conducted in Zimbabwe-- a limitation of the state of the research; not of this review. Nevertheless, there is currently no critical analysis of what this reality (all the information is from and about one country only) means for this study aim.

A major methodological shortcoming is the lack of detail regarding the process to extract and analyze barriers and facilitators. The authors included “major themes,” but how were they derived? Was there any formal coding of the barriers and facilitators?

Other specific questions/comments:

- Correct format of in-text references. For example, in the first paragraph, the first 2 references (Health 2011 and Organization 2017). Another example: line 314: "...The remaining article, Chibanda et al’s (2017) case study,(Chibanda 2017) was deemed as unable to be assessed..."

- Regarding statement in lines 62-63: “...there remains a need to understand how to implement and scale-up these interventions to address the treatment gap.”: Please elaborate on the rationale (and supporting references). What are we trying to understand about the implementation an scale up?

Methods

- Data Extraction section: Please include inclsion/exclusion criteria for the title and abstract sceening process.

- Why conducted the search again in December 2023, and what were the results? Did the screenings start after this second search (12/23) or were they repeated?

Results

- Implementation research outcomes: The current presentation of these findings is too wordy; please revise to be more concise and more easily comprehensible. Perhaps a table of articles (in rows) and implementation outcomes included (in columns), with appropriate check marks, can better illustrate these findings.

- Lines 223-227: include a bit more contextual detail about the “One study” -- for example, the “A study that implemented Friendship Bench among XXX in XXX”

- lines 281-282: change “in the RE-AIM study” to “in the study using the RE-AIM framework”

- Lines 296-307: Scalability - “Themes that emerged included..”  did you do a formal qual analysis to derive at the themes?

- Line 337: "’reach' is an accurate measurement of...“ -- what do you mean it is an ”accurate" measurement of the level of integration within a service setting?

- Facilitators: A more explicit link between facilitators and implementation outcomes would be hellpful. The authors started with the first 2 domains, but then did not mention them afterwards -- e.g.,

- patient/client characteristics  acceptability and appropriateness

- provider characteristics  acceptability

- intervention  ???

- organizational factors  ???

- Barriers: In general, this section is not as detailed as the section on facilitators; no specific mentions of which article(s) highlighted which barrier(s)

Strengths and limitations

- lines 451-453: unclear sentence (run-on); please revise

---

## [Reviewer Report]

The article is improved in some ways, but two major problematic issues remain:

1. The ‘LMIC’ framing of the article.

I understand that the search strategy included all LMICs, but because Friendship Bench (FB) has not been studied (and reported) for CMDs in any other country besides Zimbabwe, statements about the implementation (outcomes) associated with FB in LMICs are purely conjecture. The authors could include 1-2 sentences in the Discussion about this issue, but it shouldn’t be used to frame the article.

In addition, I suggest that the authors change the title to something like: ‘Evaluating implementation research outcomes for Friendship Bench: A systematic review ’

2. The discussion of barriers and facilitators.

The analysis of B/F’s and their relation to IOs is in the Discussion section. However, it is currently mixed with content that reports findings from the included studies, which should be in the Results section, and should be referenced similar to the text on Implementation Outcomes. The text in the Discussion section should include analyses and synthesis of the findings in the context of the literature. This is where the authors could include a discussion of how the findings of this review may be similar or different in other LMICs.

Additionally, since the focus (as seen by the relative length of the text) seems to be on the implementation outcomes, the authors should consider switching the description of Proctor’s IO framework before Le et al.’s BeFITS-MH framework.

---

## [Reviewer Report]

The manuscript still requires substantial changes to make it more coherent and a contribution to the literature. Mainly, the analysis and discussion of the barriers and facilitators are still very cursory, and are disjointed from that of implementation outcomes. The connection between which barriers and facilitators are typically associated with which implementation outcomes would substantially strengthen the contribution of the article. However, if these analyses are intended to be separate, the authors can clearly state so in the introduction -- i.e., The research questions of this systematic review are: (1) ...., (2) ...

DETAILED FEEDBACK:

ABSTRACT:

Line 27: “All studies...within the past 8 years and reported on all implementation research outcomes”  revise. Current wording suggests that all articles reported on all implementation outcomes, which is not correct.

INTRODUCTION:

Lines 100-101: “an understanding of its implementation on a larger scale beyond Zimbabwe is still needed”  suggest changing to: “...a comprehensive understanding of its implementation ”determinants“ (i.e., barriers and facilitators) and ”successes“ (i.e., implementation outcomes) is needed to inform scale up of the Friendship Bench in other LMICs.”; I also suggest moving this sentence to the preceding paragraph.

Lines 102-103: This sentence is the introduction for the application of the BeFITS-MH framework, which is currently split from the description of the BeFITS-MH due to the moving of Proctor’s IO description. I

Lines 132-134: This sentence should immediately follow description of Proctor’s

Lines 135-139: I suggest moving this up after lines 100-101, to introduce the objectives of the review. Then, you can describe the frameworks.

RESULTS:

Lines 226-228: instead of citing the 6 articles, only cite the 1 article that did not mention or evaluated scalability.

Lines 319-392 (entire sections on Facilitators and Barriers): Although the authors moved these sections from the Discussion as suggested, the text currently summarizes on a rather superficial level the facilitators and barriers.The mentioned factors (in each category) are not supported by the references that they came from, so the audience cannot ascertain the validity of the findings.

Lines 370-372: “Additionally, the Friendship Bench needs to be expanded to other mental and behavioral health conditions such as post-traumatic stress disorder and substance use disorder.”  It’s meant to be a low-intensity psychological intervention.

Lines 403-406: change to “This is the first systematic review to synthesize the findings on the implementation determinants and outcomes of the Friendship Bench as a task-sharing intervention to address common mental disorders.”

Table 1: The table included in this version only has the authors and whether or not each of the implementation outcomes were assessed. Please revert back to the original table, which had other characteristics. And since in lines 166-169, the authors stated that data extraction included barriers and facilitators, please include these as well.

In the old table 1, where how IO were assessed, please match the assessments with the appropriate IO, where possible. For example, in Abas et al 2016, list like so: Acceptability: number of patients and visits; Appropriateness: themes emerged from FGDs and IDIs; Scalability: workforce turnover.

---

## [Editor Report]

Can you kindly succinctly map the barriers and facilitators to implementation outcomes as suggested by the reviewer?

---

## [Reviewer Report]

Thank you for the additional revisions.

The abstract and the discussion (lines 409-410) still contain language that suggests all 7 included articles reported on all implementation research outcomes. Please revise accordingly.